# Maize Genotypes Sensitive and Tolerant to Low Phosphorus Levels Exhibit Different Transcriptome Profiles under *Talaromyces purpurogenus* Symbiosis and Low-Phosphorous Stress

**DOI:** 10.3390/ijms241511941

**Published:** 2023-07-26

**Authors:** Qing Sun, Peiyu Zhang, Zixuan Zhao, Xuefang Sun, Xiang Liu, Hongsheng Zhang, Wen Jiang

**Affiliations:** Shandong Provincial Key Laboratory of Dryland Farming Technology, College of Agronomy, Qingdao Agricultural University, Qingdao 266109, China; sunqing@qau.edu.cn (Q.S.); zhangpeiyu@stu.qau.edu.cn (P.Z.); zhaozixuan@stu.qau.edu.cn (Z.Z.); sunxuefang@qau.edu.cn (X.S.); 20222101049@stu.qau.edu.cn (X.L.); zhanghsh0812@qau.edu.cn (H.Z.)

**Keywords:** endophytic fungus, maize, genotype, RNA-seq, root–fungus symbiosis

## Abstract

*Talaromyces purpurogenus*, an endophytic fungus, exhibits beneficial effects on plants during plant–fungus interactions. However, the molecular mechanisms underlying plants’ responses to *T. purpurogenus* under low-phosphorous (P) stress are not fully understood. In this study, we investigated the transcriptomic changes in maize with low-P-sensitive (31778) and -tolerant (CCM454) genotypes under low-P stress and its symbiotic interaction with *T. purpurogenus*. Its colonization enhanced plant growth and facilitated P uptake, particularly in 31778. Transcriptome sequencing revealed that 135 DEGs from CCM454 and 389 from 31778 were identified, and that only 6 DEGs were common. This suggested that CCM454 and 31778 exhibited distinct molecular responses to *T. purpurogenus* inoculation. GO and KEGG analysis revealed that DEGs in 31778 were associated with nicotianamine biosynthesis, organic acid metabolic process, inorganic anion transport, biosynthesis of various secondary metabolites and nitrogen metabolism. In CCM454, DEGs were associated with anthocyanin biosynthesis, diterpenoid biosynthesis and metabolic process. After *T. purpurogenus* inoculation, the genes associated with phosphate transporter, phosphatase, peroxidase and high-affinity nitrate transporter were upregulated in 31778, whereas AP2-EREBP-transcription factors were detected at significantly higher levels in CCM454. This study provided insights on the molecular mechanisms underlying plant–endophytic fungus symbiosis and low-P stress in maize with low-P-sensitive and -tolerant genotypes.

## 1. Introduction

Phosphorus (P) is an essential nutrient for plants and plays a crucial role in many functions, such as anabolism and catabolism. It is one of the most important fertilizers for high-yielding crops [1]. However, its slow diffusion in soil causes its deficiency, which is a common problem in agro-ecosystem lands [2]. Due to the precipitation and mineralization of inorganic phosphate, crops can only utilize 15–25% of applied P fertilizer. Hence, approximately 70% of arable land suffers from suboptimal phosphate availability for vegetative growth, hampering crop productivity [3,4]. Furthermore, P fertilizers must be prepared from phosphates from rock, which is a finite resource. Therefore, at current consumption rates, a scarcity of P fertilizers is going to be a real crisis by the end of this century [5]. Plants have evolved a series of developmental and physiological responses to low-P stress, including secreting acid phosphatases, releasing organic acids, inducing Pi transporters with high affinity and changing the shoot–root biomass ratio [6]. 

The association between plants and beneficial fungi provides several benefits to plants, including increased biomass production [7], enhanced nutrient uptake [8,9,10] and environmental adaptation [11]. The inoculation of endophytes in the rhizosphere is considered one of the most sustainable and promising approaches to promote plant growth as it significantly increases the efficiency of nutrient absorption and improves soil functions (i.e., increases soil nutrients, balances pH and increases soil microbial diversity), and is beneficial for the environment as it reduces artificial fertilizer input [12,13]. The colonized endophytes help their plant hosts in adapting to adverse soil conditions and promoting growth through several mechanisms [14,15]. The inoculation of endophytes significantly improves the germination rate, seedling survival rate and biomass of plants [12,16]. Xu et al. reported that through enhancing phosphorus-metabolism-related enzyme activities and decreasing the pH of the rhizosphere, *Exophiala pisciphila* H93 inoculation increased biomass, P absorption and photosynthesis in maize compared with uninoculated maize plants [17]. Sun et al. reported that *Falciphora oryzae*, an endophytic fungus, increased the lateral root growth of *Arabidopsis* by producing indole derivatives [18]. 

The molecular interactions between symbiotic fungi and plants were recently clarified, especially those related to P acquisition induced with arbuscular mycorrhizal fungi (AMF). *PHR2*, a transcriptional regulator of the response to phosphate starvation in rice, is involved in mycorrhizal root colonization, mycorrhizal phosphate uptake and yield increase in field soils [19]. In rice, *NSP2* and *ZAS* promote root colonization; *CERK1* and *SYMRK* play a role in signal transduction at the root epidermal surface; and *PT11* plays a role in nutrient uptake [20]. The phosphate transporter (*PHT*) gene family is involved in phosphate transport. In maize, *ZmPHT1;2*, *ZmPHT1;4*, *ZmPHT1;6*, *ZmPHT1;7*, *ZmPHT1;9* and *ZmPHT1;11* were upregulated by AMF, suggesting that these genes might participate in AMF-induced P absorption and/or transport [21]. These studies revealed important molecular mechanisms underlying the establishment of plant–fungus symbioses and provided molecular targets for gene editing to establish symbiotic relationships between plants and beneficial mycorrhiza. However, in plants, the regulation of P uptake and P-transport genes induced by fungi other than AMF have not been studied in detail.

Maize is an important staple cereal that contributes to human food and animal feed worldwide [22]. The maize yield is frequently negatively affected by various abiotic stresses, particularly low-P stress in tropical and subtropical soils [23]. The edaphic properties and genetic characteristics of plants determine their efficiency of P acquisition [24]. In maize with genotypes of greater or sustained development of the lateral roots, the ability to acquire P is enhanced compared with other genotypes [25]. Plant growth promoting rhizobacteria and mycorrhizae can also mediate P availability to plants. A new maize genotype (XY335) exhibited significantly enhanced phosphorus acquisition efficiency (PAE) compared with an old genotype (HMY); this was attributed to the stronger mycorrhizal responsiveness of the new genotype [26]. Rhizosphere microbes play a crucial role in plant health by promoting nutrient acquisition, immune modulation and abiotic stress tolerance. Therefore, the interaction between plants and microbes is a key factor for determining global crop production and ecosystem stability [27,28]. However, the effects of endophytic fungi on maize with genotypes of varying P sensitivity have not been explored in detail.

*Talaromyces purpurogenus* is a plant endophyte that produces terpenoids, antiproliferative red pigments and antioxidative bioactive compounds [29,30]. Interestingly, *T. purpurogenus* can solubilize insoluble inorganic P and mineralize organic P with a strong ability to regulate the nutrient dynamics in plants [31]. In this study, we aimed to comprehensively analyze the gene regulatory mechanisms in different genotypes of maize during plant–*T. purpurogenus* symbiosis under low-P stress. The growth and P content of maize seedlings exposed to low-P stress in the presence and absence of *T. purpurogenus* were assessed. The effect of *T. purpurogenus* on the molecular responses of maize seedlings under low-P stress was analyzed via transcriptome sequencing, and the differentially expressed genes (DEGs) were annotated using Gene Ontology (GO) and Kyoto Encyclopedia of Genes and Genomes (KEGG) enrichment pathway analyses to reveal the pathways involved. We hypothesized that *T. purpurogenus* could colonize the root system of maize, and the response of maize to endophytic fungi would be different with different genotypes under low-P stress. This study provided insights on the molecular regulatory mechanisms in plants during plant–endophytic fungi symbiosis under low-P stress. 

## 2. Results

### 2.1. T. purpurogenus Improves the Adaptation of Maize to Low-P Stress

To investigate the impact of *T. purpurogenus* on the adaptation of maize seedlings to low-P stress, the plants were cultivated on vermiculite supplemented with calcium phytate as an insoluble phosphate source with or without *T. purpurogenus*. As expected, *T. purpurogenus* was detected in the maize roots after their inoculation, and no fungal symbiosis was observed in the roots of the uninoculated control (Appendix A).

The growth of maize after 15 days of fungal colonization in pot cultures is shown in Figure 1A. Compared with the control group, *T. purpurogenus* significantly promoted the growth of 31778 (low-P-sensitive inbred line), whereas no significant difference was observed in the growth of CCM454 (low-P-tolerant inbred line). Compared with the control group, *T. purpurogenus* inoculation significantly increased the fresh weight of the shoot by 55.7% and the total plant P content by 32.9% in 31778 (*p* < 0.05). However, it only significantly increased the total P content of the shoot by 22.4% (*p* < 0.05) and the fresh weight of the shoot remain unchanged in CCM454 (Figure 1B, C). Moreover, *T. purpurogenus* improved root morphological traits, such as root length and root biomass, in 31778; however, no significant change was observed in CCM454 (Figure 1D). *T. purpurogenus* significantly increased the fresh weight of the root by 30.9% in 31778; however, no change was observed in CCM454 (Figure 1E). *T. purpurogenus* significantly increased the total P content of the root by 9.8% in 31778 and 3.1% in CCM454 (Figure 1F).

### 2.2. Quality Assessment of Transcriptome Sequencing

To systematically identify mRNAs that respond to *T. purpurogenus* under low-P stress, the transcriptome of maize seedling roots was sequenced after 15 days of the fungal inoculation. On average, 48,778,439 raw reads were generated per sample (Appendix A). After deleting adapter and low-quality reads, 540,555,118 clean and high-quality reads were obtained. Sequence alignment was performed using the B73 genome (RefGen_v4.41) as the reference genome. Among the 12 samples, the mapping ratios varied between 89.29% and 90.35%, indicating that the majority of reads were mapped to the B73 genome, and the selected reference genome could meet the analysis requirements (Table 1).

### 2.3. Analysis of Differentially Expressed Genes (DEGs) in Low-P-Tolerant and -Sensitive Genotypes of Maize

The DEGs between *T. purpurogenus*-inoculated and -uninoculated samples were determined using pair-wise comparisons (Figure 2). Lines 31778 and CCM454 exhibited 389 and 135 DEGs, respectively, in response to *T. purpurogenus*. In 31778 and CCM454, 173 and 55 DEGs were upregulated and 216 and 80 DEGs were downregulated, respectively. The number of fungus-responsive genes was lower in CCM454 than 31778, indicating that 31778 exhibited a greater response to *T. purpurogenus* than CCM454. Moreover, only six DEGs were common between CCM454 and 31778 after the treatment with *T. purpurogenus* (Figure 3).

### 2.4. GO and KEGG Pathway Enrichment Analyses of the DEGs

GO functional enrichment analysis was conducted to annotate the functions of DEGs. For 31778, the DEGs in *T. purpurogenus*-inoculated and -uninoculated samples were enriched in 556, 72 and 294 terms in the biological process (BP), cellular component (CC) and molecular function (MF), respectively (Appendix A). The significantly enriched BP terms were the nicotianamine metabolic process, nicotianamine biosynthetic process, organic acid metabolic process and inorganic anion transport (Figure 4A). The significantly enriched MF terms were nicotianamine synthase activity and oxidoreductase activity. In the KEGG pathway enrichment analysis, these DEGs were clustered in 52 pathways (Appendix A), the most significant being biosynthesis of various secondary metabolites, nitrogen metabolism and cysteine and methionine metabolism (Figure 5A). For CCM454, the DEGs in *T. purpurogenus*-inoculated and -uninoculated samples were enriched in the anthocyanin-containing compound biosynthetic process, anthocyanin-containing compound metabolic process, pigment biosynthetic process and pigment metabolic process (BP terms); and transcription regulator activity and DNA-binding transcription factor activity (MF terms) in the GO analysis (Appendix A, Figure 4B). In the KEGG pathway enrichment analysis, these DEGs were enriched in 31 pathways (Appendix A), the most significant being anthocyanin biosynthesis, diterpenoid biosynthesis and the plant–pathogen interaction (Figure 5B).

### 2.5. Comparison of the DEGs Induced by T. purpurogenus in Low-P-Sensitive and -Tolerant Genotypes of Maize

The molecular response of low-P-sensitive and -tolerant genotypes of maize to *T. purpurogenus* under low-P stress was assessed by identifying the DEGs induced by *T. purpurogenus* (Table 2). After *T. purpurogenus* inoculation, one gene each of P transporter *Zmpht1;9* (*Zm00001d027700*), trehalose-6-phosphate phosphatase9 (*Zm00001d022192*), calcium-dependent protein kinase29 (*Zm00001d052713*) and MYB-transcription factor 114 (*Zm00001d011739*) was significantly upregulated in 31778 but exhibited no difference in CCM454. Four peroxidase genes (*Zm00001d034128*, *Zm00001d014603*, *Zm00001d026683*, and *Zm00001d002901*) and four high-affinity nitrate transporter genes (*Zm00001d054060*, *Zm00001d054057*, *Zm00001d011679*, and *Zm00001d017095*) were significantly upregulated in 31778 but not in CCM454. In CCM454, two AP2-EREBP-transcription factor genes (*Zm00001d010175* and *Zm00001d039019*) were upregulated and one AP2-EREBP-transcription factor gene (*Zm00001d049364*) was downregulated but exhibited no difference in 31778. One WRKY-transcription factor (*Zm00001d020492*) was downregulated in CCM454 but not in 31778. 

### 2.6. Validation of the DEGs Using RT-qPCR

To validate the outcomes of the high-throughput sequencing, eight DEGs (*Zm00001d027700*, *Zm00001d052713*, *Zm00001d023617* and *Zm00001d025409* from 31778; *Zm00001d010175*, *Zm00001d039019*, *Zm00001d049364* and *Zm00001d044558* from CCM454) involved in the *T. purpurogenus*–maize root interaction were randomly selected for RT-qPCR analysis. Consistent with high-throughput sequencing results, four DEGs were significantly upregulated and four were downregulated (Figure 6), demonstrating the reliability of the sequencing results.

## 3. Discussion

Plants can form symbiotic associations with rhizosphere microorganisms. Endophytic fungi can colonize hundreds of plant species and contribute to nutrition acquisition and stress resistance in plants. However, molecular mechanisms governing the interactions between soil endophytes and their plant hosts are still poorly understood. Here, we studied the impact of maize–endophytic fungus symbiosis on the transcriptome of the maize root with low-P-sensitive and -tolerant genotypes under low-P stress. *T. purpurogenus*, an endophytic fungus isolated from maize rhizosphere soil, efficiently solubilizes insoluble organic and inorganic P. In this study, a *T. purpurogenus*–maize symbiotic system was used to explore the molecular mechanisms impacted by the interaction.

### 3.1. Promotion of Maize Growth by T. purpurogenus

The colonization of maize roots by *T. purpurogenus* resulted in the development of microsclerotia-like structures within the root cells (Appendix A). The formation of such structures commonly occurs in the symbiotic associations of endophytic fungus–root, as evidenced by previous reports on the colonization of maize roots by *Serendipita indica*, an endophytic fungus [32]. After inoculating *E. pisciphila*, an endophytic fungus, hyphae and microsclerotia were observed in all maize root samples but not in the uninoculated control [17]. Our findings suggested that *T. purpurogenus* could develop an endophytic phase and possessed a specific interface that may facilitate the growth of host plants.

Symbiotic fungi can greatly improve the host plant’s growth and ability to absorb nutrients. *E. pisciphila* enhanced the biomass, P absorption and photosynthesis in maize under both P-deficient and -sufficient conditions [17]. *F. oryzae* could promote lateral root growth and development in *Arabidopsis thaliana* [18]. *S. indica* could greatly promote nutrient uptake and improve maize growth in low-sulfate environments [32]. Similarly, in this study, *T. purpurogenus* promoted maize root growth and P uptake, and the growth-promoting effects were dependent on the plant genotype. These findings suggested that *T. purpurogenus* promoted maize root growth through a genotype-dependent mechanism.

### 3.2. Alteration in the Maize Transcriptome by T. purpurogenus

The molecular mechanisms of the contribution of the endophytic fungi–plant association to the growth and P absorption in maize were explored via transcriptome sequencing of the maize seedling roots with and without *T. purpurogenus* inoculation. Very few common DEGs induced by *T. purpurogenus* were observed between 31778 and CCM454. This indicated that in the presence of endophytic fungi, the molecular mechanisms of maize to enhance P utilization vary depending on the genotype. 

The most enriched GO terms were the nicotianamine metabolic process, nicotianamine biosynthetic process, organic acid metabolic process, oxidoreductase activity and inorganic anion transport for the DEGs in 31778; and the anthocyanin-containing compound biosynthetic process, anthocyanin-containing compound metabolic process, DNA-binding transcription factor activity and transcription regulator activity for the DEGs in CCM454. A previous study reported that nicotianamine is a non-secretory secondary metabolic substance produced by plants and that it functions as a potential antioxidant in plant resistance [33]. Similarly, our results indicated that *T. purpurogenus* could increase the adaptability of maize to low-P stress by regulating the biosynthetic and metabolic process of nicotinamine. To increase P acquisition under P-deficient conditions, roots can produce organic acid and secrete it into the rhizosphere as an adaptive strategy. A higher organic acid exudation in P-starved plants has been observed, particularly in maize with low-P-tolerant genotype [34]. This was consistent with our results, which indicated that the organic acid metabolic process significantly changed in the low-P-sensitive inbred line 31778. In the low-P-tolerant inbred line CCM454, the anthocyanin-containing compound biosynthetic and metabolic process was strongly altered. Anthocyanins participate in the defense response of plants to abiotic stress [35,36] and mediate certain plant–microbe interactions [37]. Pei et al. reported that anthocyanins improved root growth, reduced oxidative destruction and enhanced photosynthetic performance in maize under low-P stress [38]. In this study, *T. purpurogenus* could enhance the ability of maize to resist P stress by regulating the anabolism of anthocyanin in the low-P-tolerant inbred line CCM454.

After *T. purpurogenus* inoculation, strongly enriched KEGG pathways included the biosynthesis of various secondary metabolites, nitrogen metabolism and cysteine and methionine metabolism in 31778; and anthocyanin biosynthesis, diterpenoid biosynthesis, and plant–pathogen interactions in CCM454, indicating that *T. purpurogenus* altered different pathways in different genotypes of maize.

### 3.3. Effect of T. purpurogenus on the Genes Involved in P acquisition under Low-P Stress

PHTs play an important role in phosphate uptake in plants. *ZmPht1;9* is a mycorrhiza-induced phosphate transporter and plays a significant role in phosphate uptake and plant growth [39]. The expression level of *ZmPht1;6* was 26–135 times higher and that of *ZmPht1;3* was 5–44 times lower in maize roots with AMF than in those without [40]. In this study, the upregulated expression of *ZmPht1;9* in 31778 inoculated with *T. purpurogenus* was consistent with the role of *T. purpurogenus* in facilitating the uptake of P under low-P conditions. These results suggested that *ZmPHTs* can be further exploited for improving phosphate uptake in crop growth.

In this study, four peroxidase genes were significantly upregulated in 31778 inoculated with *T. purpurogenus*. Peroxidase family genes are closely related to controlling ROS homeostasis in plant roots [41]. Peroxidase activity in the roots of 31778 and CCM454 under P-deficient conditions was higher than under P-sufficient conditions [42]. Our results confirmed that genes related to the defense response were more highly expressed in the roots inoculated with endophytic fungi, suggesting an interaction between endophytic fungi or phosphate stress and plant immunity. Our findings supported the suggestion that peroxidase could be an important trait in maize stress resistance breeding.

Studies have demonstrated that altering the expression of a transcription factor can influence the resistance to P stress by activating downstream target genes. The transcription factors include members of the MYB, WRKY and AP2/EREBP families [43]. In this study, the fungal-responsive DEGs in 31778 and CCM454 included one MYB, three WRKYs and three AP2/EREBP transcription factors. *PHR2*, a MYB transcription factor responsible for phosphate starvation responses in rice, is required for root colonization, mycorrhizal phosphate uptake and yield enhancement in field soil [19,20]. WRKY transcription factor controls several co-expressed genes, while the overexpression of *OsWRKY74* increases the expression of Pi transport proteins such as *OsPHT1;3*, *OsPHT1;4* and *OsPHT1;10* [44]. AP2/EREBP transcription factor *WRI5a* in *Medicago truncatula* is a master regulator of AMF symbiosis controlling lipid transfer and periarbuscular membrane formation [45]. These results indicated that transcriptional regulation plays a crucial role in the interaction between *T. purpurogenus* and maize under low-P conditions. It is expected that the transcription factors will greatly enrich the available gene resources toward increasing low-P tolerance in maize.

Interestingly, four high-affinity nitrate transporter genes were significantly upregulated in 31778 inoculated with *T. purpurogenus*. To maintain a nutrient balance and achieve optimal growth, nitrogen (N) and P uptake and metabolism must be coordinated in plants [46]. Reduced N uptake, translocation and assimilation are the typical effects of P starvation on the N response [47]. Medici et al. reported that P starvation negatively regulates *NRT1.1* transcription and protein stability [48]. In the current study, nitrate transporter genes were significantly upregulated in 31778, implying that N absorption was increased after *T. purpurogenus* inoculation.

In addition, one trehalose-6-phosphate phosphatase9 gene and one calcium-dependent protein kinase29 gene were significantly upregulated in 31778. Trehalose-6-phosphate phosphatase induces ROS production during ABA-controlled root elongation and stomatal movement process [49]. In *Arabidopsis*, calcium-dependent protein kinase 29 modulates the polarity of PIN-FORMED and root development via its own phosphorylation code [50]. These results indicated that different molecular pathways are involved in the root–fungus symbiosis under low-P stress depending on the maize genotype. Therefore, these identified genes are promising candidates for breeding maize cultivars with improved low-P tolerance.

## 4. Materials and Methods

### 4.1. Plant Material and Growth Conditions

Maize seeds of low-P-sensitive (31778) and -tolerant (CCM454) genotypes were obtained from the Maize Molecular Breeding Laboratory, Institute of Crop Sciences, Chinese Academy of Agricultural Sciences. The criteria for distinguishing between low-P sensitive and low-P tolerant genotypes have been described in a previous article published by our laboratory [42]. After surface sterilization with 3% NaClO for 20 min, the seeds were washed three times in distilled water and allowed to germinate on moist filter paper in humidified sterile Petri dishes for 2–3 days at 28 °C. Uniformly germinated seeds were cultivated in plastic pots with air-dried vermiculite. Calcium phytate was supplied as an insoluble P source. There were four treatments in this study, including 31778 with and without inoculation, and CCM454 with and without inoculation. In each pot of the inoculated group, 1 mL of 10^7^ CFU/mL *T. purpurogenus* spore suspension was added to the vermiculite, and in the control, an equal amount of sterile water was added. The low-P stress seedlings were irrigated with Hoagland’s nutrient solution without KH_2_PO_4_ every 3 days. The seedlings were grown in a greenhouse at 28 °C/22 °C with a day/night cycle of 14 h/10 h. The relative humidity was 45–55%. Fifteen days after the fungal inoculation, six maize roots and shoots were collected for physiological and transcriptome analyses. 

### 4.2. Histochemical Analysis Using Trypan Blue Staining

Three maize root samples were randomly collected from *T. purpurogenus*-inoculated and -uninoculated plants. The roots were softened with 10% KOH solution for 15 min, acidified with 1 N HCl for 10 min and stained with 0.02% Trypan Blue. After 2 h, the samples were destained with 50% lactophenol for 1–2 h and observed under a Leica light microscope (Leica, Wetzlar, Germany). 

### 4.3. Determination of Biomass and P concentration

From each treatment, six plants were randomly selected. The fresh weights of roots and shoots were measured using an electric balance. The shoots and roots were dried at 105 °C for 30 min, followed by 65 °C for 72 h. After complete drying, the shoots and roots from each replicate were ground to a fine powder to determine the plant’s total P content. In total, 0.2 g powder was digested with H_2_SO_4_–H_2_O_2_, and P content was determined using vanadomolybdate method [26].

### 4.4. RNA Extraction and Sequencing

Total RNA was extracted from the roots using TransZol Plant Reagent (from TransGen Biotech, Beijing, China) as per the manufacturer’s instructions. The quantity and quality of the extracted RNA were determined using the NanoDrop one C (Thermo Fisher Scientific, Waltham, MA, USA) and 1% agarose gel electrophoresis, respectively. RNAseq transcriptome library was prepared with NEB Next Ultra II RNA Library Prep Kit for Illumina (NEB, Houston, TX, USA) using 1 μg of total RNA. First, the mRNA was purified via poly-T selection method using oligo(dT) beads, and the fragments were prepared using divalent cations in a fragmentation buffer under elevated temperatures. Further, double-stranded cDNA was synthesized using random oligonucleotides and Super Script II. The synthesized cDNA was purified, followed by adenylation of 3′ ends and ‘A’ base addition. The libraries were optimized for cDNA fragments of 400–500-bp length, and PCR amplification was performed using Illumina PCR Primer Cocktail for 15 cycles. After quantification on an Agilent high sensitivity DNA Bioanalyzer 2100, paired-end RNA-seq libraries were sequenced by Shanghai Personal Biotechnology Cp. Ltd. using the NovaSeq 6000 platform (Illumina, San Diego, CA, USA). The RNA sequencing data are available through the NCBI Sequence Read Archive (SRA; accession number PRJNA986686).

### 4.5. Mapping of RNA-seq Reads and Identification of DEGs

To obtain clean reads with high-quality sequence, raw reads were trimmed and quality-controlled with Cutadapt (v1.15) and FastQC (v0.11.8). The clean reads were separately mapped to the maize B73 RefGeen_V4 genome with orientation mode using HISAT2 (v2.0.5) software [51]. Using HTSeq (0.9.1) statistics, the expression level of each transcript was calculated based on fragments per kilobases per million fragments (FPKM) method to identify DEGs between various samples. DEGs with |log_2_FoldChange| > 2 and *p* < 0.05 (DESeq 1.30.0) were considered significant DEGs.

### 4.6. GO and KEGG Pathway Enrichment Analyses

GO analysis was conducted using topGO to identify functionally differentially enriched genes by calculating *p* values based on hypergeometric distributions (http://geneontology.org/ (accessed on 8 January 2022)). KEGG pathway enrichment analysis was conducted to analyze metabolic pathways of DEGs using ClusterProfiler (3.4.4) software (http://www.kegg.jp/ (accessed on 16 January 2022)). Enriched GO and KEGG categories with *p* < 0.05 were considered significant.

### 4.7. RT-qPCR

The first-strand cDNA was synthesized using SuperScript TM III First-Strand Synthesis Supermix (Invitrogen, Waltham, USA). The primers are given in Appendix A. RT-PCR was performed on an ABI 7500 system (Thermo Fisher Scientific, Waltham, MA, USA) with the SYBR Premix Ex Taq^TM^ kit (TaKaRa, Osaka, Japan). The PCR reaction system and conditions were as described by Zhao et al. [52]. All reactions were performed in triplicate. Actin gene was used as the reference gene. The relative expressions were calculated according to the 2^−∆∆Ct^ method.

### 4.8. Statistical Analyses

All experiments were carried out at least three times. One-way ANOVA was performed using SPSS 23.0 (SPSS Inc., Chicago, IL, USA) for statistical analysis. Duncan’s post hoc test was used to perform multiple comparisons and *p* < 0.05 was considered significant.

## 5. Conclusions

In summary, in this study, the different phenotypes and molecular mechanisms of two contrasting maize genotypes were described in the presence of a root endophyte under low-P stress (Figure 7). The symbiotic relationship between *T. purpurogenus* and maize roots enhanced low-P stress tolerance as evidenced by improved plant growth, root architecture and P uptake, particularly for the low-P-sensitive inbred line 31778. Transcriptome analysis indicated that line 31778 exhibited greater molecular responsiveness to *T. purpurogenus* than CCM454 under low-P stress. Furthermore, 31778 enriched more DEGs in nicotianamine biosynthesis, organic acid metabolic process, inorganic anion transport, biosynthesis of various secondary metabolites and nitrogen metabolism.CCM454 enriched more DEGs in anthocyanin biosynthesis, the metabolic process and diterpenoid biosynthesis. Moreover, one phosphate transporter gene, four peroxidase genes, four high-affinity nitrate transporter genes and four transcription factor genes may play an important role in regulating the process of the plant–fungus interactions. Our study provided novel insights on the molecular mechanisms underlying plant–endophytic fungus interactions under low-P stress. Further studies should focus on the characterization of P assimilation gene-expression profiles and how they relate to the P balance and phosphate demand for both organisms.

## Figures and Tables

**Figure 1 ijms-24-11941-f001:**
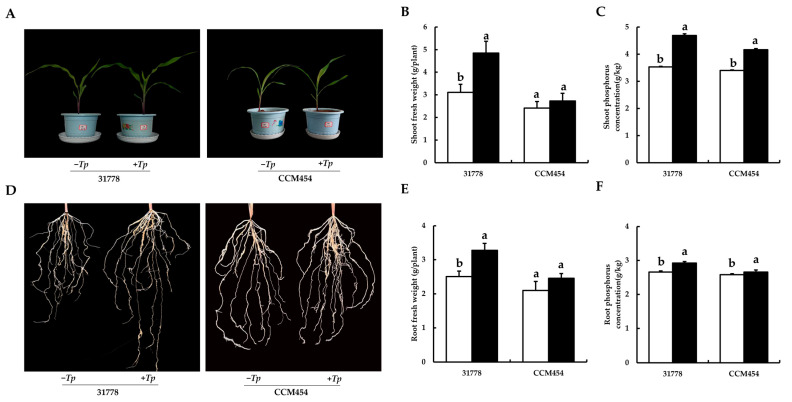
Physiological responses of maize seedlings to *T. purpurogenus* under low-P stress. (**A**) Morphological appearance of shoot. (**B**) The fresh weight of shoot. (**C**) Total P content of shoot. (**D**) Morphological appearance of the root. (**E**) The fresh weight of root. (**F**) Total P content of root. The black and white bars indicate *T. purpurogenus*-inoculated and -uninoculated plants, respectively. Values indicate mean ± SD, *n* = 6. Various lowercase letters indicate significant difference at *p* < 0.05.

**Figure 2 ijms-24-11941-f002:**
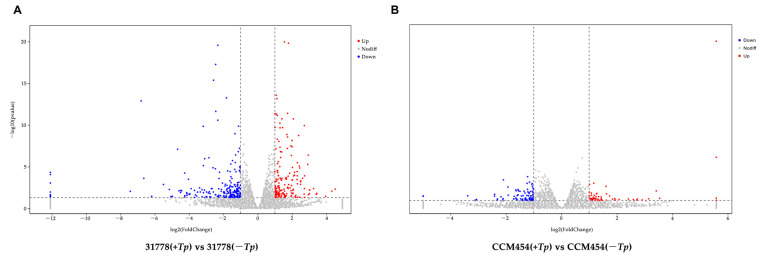
Volcano plots illustrating the up- and downregulated DEGs in low-P-sensitive and -tolerant genotypes of maize. (**A**) Low-P-sensitive and (**B**) low-P-tolerant genotypes. Red and blue colors indicate up- and downregulated DEGs, respectively, based on the log2 fold change and *p* < 0.05.

**Figure 3 ijms-24-11941-f003:**
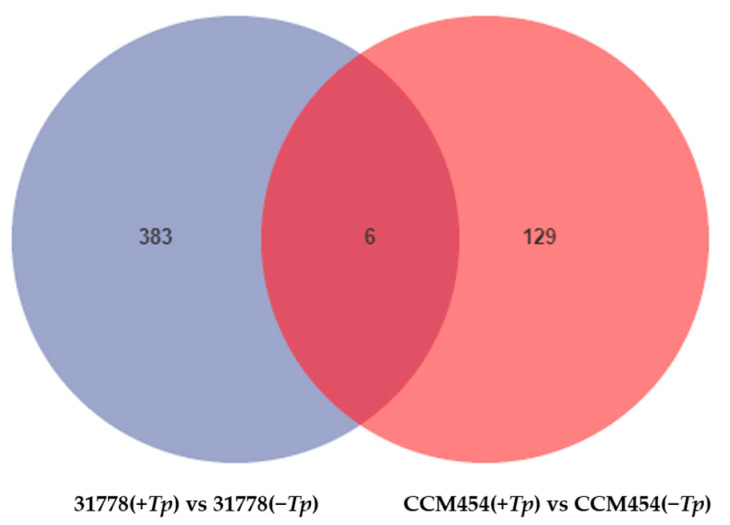
Venn diagram showing the up- and downregulated DEGs in low-P-sensitive and -tolerant genotypes of maize. DEGs were quantified based on log2 fold changes ± 2 and FDR < 0.05.

**Figure 4 ijms-24-11941-f004:**
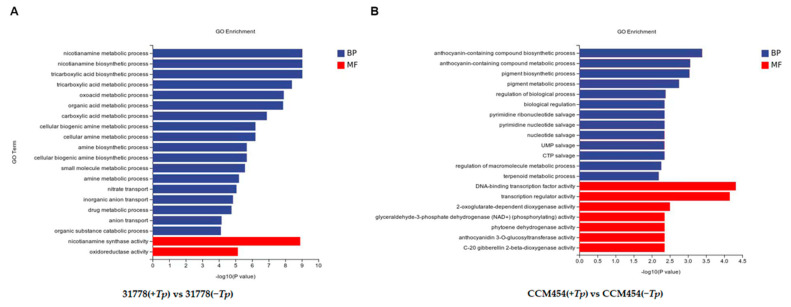
Gene Ontology analysis of differentially expressed genes (DEGs). (**A**) Low-P-sensitive and (**B**) -tolerant genotypes of maize.

**Figure 5 ijms-24-11941-f005:**
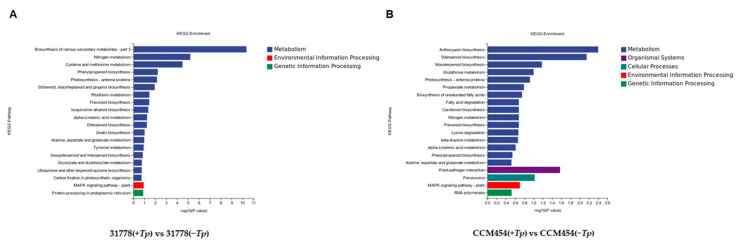
KEGG pathway enrichment analysis of DEGs. (**A**) Low-P-sensitive and (**B**) -tolerant genotypes of maize.

**Figure 6 ijms-24-11941-f006:**
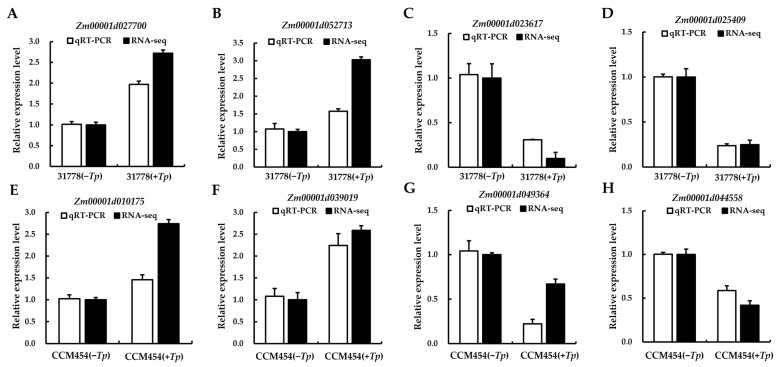
Expression of selected DEGs in response to *T. purpurogenus* using qRT-PCR. (**A**–**D**) Differences in expression levels of 31778. (**E**–**H**) Differences in expression levels of CCM454. Data are the mean ± SD of three biological replicates. The maize UBQ gene was used as an internal control for normalization of gene expression.

**Figure 7 ijms-24-11941-f007:**
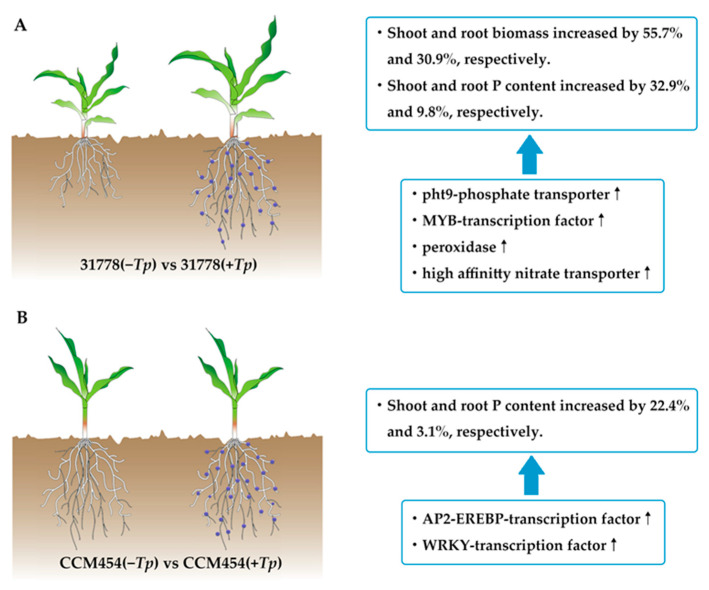
A conceptual paradigm explaining the regulatory mechanisms in different genotypes of maize during plant–*T. purpurogenus* symbiosis under low-P stress. (**A**) Low-P-sensitive and (**B**) -tolerant genotypes of maize.

**Table 1 ijms-24-11941-t001:** The comparison of the sequencing data of 12 samples with the reference genome.

Sample Name	Clean_Reads	Total_Mapped	Mapped_to_Gene	Mapped_to_Exon
31778(−*Tp*)-1	46298450	41626754 (89.91%)	38990457 (96.58%)	38557878 (98.89%)
31778(−*Tp*)-2	38782674	34628066 (89.29%)	32343885 (96.50%)	31964851 (98.83%)
31778(−*Tp*)-3	48591292	43401145 (89.32%)	40555256 (96.49%)	40064582 (98.79%)
31778(+*Tp*)-1	46370420	41568979 (89.65%)	38856443 (96.40%)	38377630 (98.77%)
31778(+*Tp*)-2	43268318	38856759 (89.80%)	36309531 (96.43%)	35953229 (99.02%)
31778(+*Tp*)-3	48115260	42470081 (88.27%)	39716251 (96.44%)	39269182 (98.87%)
CCM454(−*Tp*)-1	43400434	38847008 (89.51%)	36347742 (96.61%)	35956252 (98.92%)
CCM454(−*Tp*)-2	47374076	42504688 (89.72%)	39766918 (96.73%)	39373252 (99.01%)
CCM454(−*Tp*)-3	50215620	45146328 (89.90%)	42298870 (96.72%)	41889645 (99.03%)
CCM454(+*Tp*)-1	39743450	35907876 (90.35%)	33630363 (96.67%)	33296449 (99.01%)
CCM454(+*Tp*)-2	43849872	39553863 (90.20%)	36958797 (96.38%)	36508229 (98.78%)
CCM454(+*Tp*)-3	44545252	39963821 (89.72%)	37194552 (96.28%)	36766865 (98.85%)

**Table 2 ijms-24-11941-t002:** Differentially expressed genes involved in the interaction between *Talaromyces purpurogenus* and maize root.

Genotype	Gene ID	Regulation	Log_2_FC	Description
31778	*Zm00001d027700*	up	2.72	pht9-phosphate transporter protein9
	*Zm00001d022192*	up	1.29	trehalose-6-phosphate phosphatase9
	*Zm00001d052713*	up	3.03	calcium-dependent protein kinase29
	*Zm00001d011739*	up	2.92	MYB-transcription factor 114
	*Zm00001d034128*	up	1.74	peroxidase 73
	*Zm00001d014603*	up	1.57	peroxidase 45
	*Zm00001d026683*	up	1.42	peroxidase 12 precursor
	*Zm00001d002901*	up	1.07	peroxidase 12 precursor
	*Zm00001d054060*	up	1.40	high affinity nitrate transporter
	*Zm00001d054057*	up	1.16	high affinity nitrate transporter
	*Zm00001d011679*	up	1.14	high affinity nitrate transporter
	*Zm00001d017095*	up	1.02	high affinity nitrate transporter
CCM454	*Zm00001d010175*	up	2.74	AP2-EREBP-transcription factor 113
	*Zm00001d039019*	up	2.59	AP2-EREBP-transcription factor 165
	*Zm00001d049364*	down	1.49	AP2-EREBP-transcription factor 209
	*Zm00001d020492*	down	1.55	WRKY-transcription factor 53

## Data Availability

Not applicable.

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
