# Peer review of "Maize Genotypes Sensitive and Tolerant to Low Phosphorus Levels Exhibit Different Transcriptome Profiles under Talaromyces purpurogenus Symbiosis and Low-Phosphorous Stress"

_ijms, 2023, doi:10.3390/ijms241511941_

Round 1

Reviewer 1 Report

In Fig. 5. (GO analysis of DEGs) the BP and MF associated genes are marked with reversed colors and in reversed order for 31778 and CCM454, respectively. This is not a fault but can be misleading for the reader.

Author Response

Dear Reviewer:

Thanks for the quick handling of our manuscript and we are pleased with the received comments. We have made corrections which we hope meet with your approval and the revisions in the manuscript have been highlighted.

In Fig. 5. (GO analysis of DEGs) the BP and MF associated genes are marked with reversed colors and in reversed order for 31778 and CCM454, respectively. This is not a fault but can be misleading for the reader.

Reply: Thank you for your suggestion. As suggested, we have modified Figure 5 and Figure 6 in the revised manuscript.

Thanks again for your comments and suggestions.

Best wishes!

Reviewer 2 Report

The manuscript of Sun et al. reports the transcriptome profiles of low-P sensitive and tolerant maize genotypes under symbiosis and low-P stress. Even though the novelty of the study is limited to only maize plant, I think it’s a good addition to the literature. The study has generated a lot of data, and the manuscript has been written and presented very well. That being said there are some minor issues that need to be addressed to improve the quality of the publication-

1.     There is discrepancy in the threshold of log2fold change determining DEGs between M&M and Results sections. In page#5, the caption of Figure 4 says the DEGs were quantified based on log2fold change ±2 (which I believe is correct), but in the M&M it says ±1(Page#11, L#388). Please correct the typo to make it consistent.

2.     The caption of the figures should be self-explanatory. It’s very difficult to understand from the Figure 1 what the authors are trying to show under +Tp condition. I would suggest authors to put more information in the captions, for example, it developed microsclerotia-like structures as a result of colonization of maize roots by T. purpurogenus and label it with an arrow in the figure to make it more understandable.

3.     Please also include in the caption or in the legend of Fig 2, what the white and black bars represent in figures B, C E and F.

4. Abstract: in the line#28, it says “were enriched in and diterpenoid biosynthesis” , it looks like the authors are either missing words (in highlighted portion) or “and” should be deleted.

I would suggest accepting the paper with minor revision.

Author Response

Dear Reviewer:

We are appreciated for your reply. The comments are all valuable and very helpful for improving our paper. As suggested, we have made corrections which we hope meet with your approval and the revisions in the manuscript have been highlighted. The main corrections in the paper and the responds to the comments are as following:

The manuscript of Sun et al. reports the transcriptome profiles of low-P sensitive and tolerant maize genotypes under symbiosis and low-P stress. Even though the novelty of the study is limited to only maize plant, I think it’s a good addition to the literature. The study has generated a lot of data, and the manuscript has been written and presented very well. That being said there are some minor issues that need to be addressed to improve the quality of the publication.

1. There is discrepancy in the threshold of log2fold change determining DEGs between M&M and Results sections. In page#5, the caption of Figure 4 says the DEGs were quantified based on log2fold change ±2 (which I believe is correct), but in the M&M it says ±1(Page#11, L#388). Please correct the typo to make it consistent.

Reply: Sorry for this kind of mistake. The DEGs were quantified based on log2fold change ±2. We have corrected the typo to make it consistent.

2. The caption of the figures should be self-explanatory. It’s very difficult to understand from the Figure 1 what the authors are trying to show under +Tp condition. I would suggest authors to put more information in the captions, for example, it developed microsclerotia-like structures as a result of colonization of maize roots by T. purpurogenus and label it with an arrow in the figure to make it more understandable.

Reply: Thank you for your suggestion. As suggested, we have put more information in the captions in Supplemental Figure S1.

3. Please also include in the caption or in the legend of Fig 2, what the white and black bars represent in figures B, C E and F.

Reply: Thanks for your suggestion. As suggested, we have added the corresponding description of the bars in the figure captions.

4. Abstract: in the line#28, it says “were enriched in and diterpenoid biosynthesis”, it looks like the authors are either missing words (in highlighted portion) or “and” should be deleted.

Reply: Sorry for this kind of mistake. We have added the missing words.

Thank again for your comments and suggestions.

Best wishes!

Reviewer 3 Report

Please see the attached review report to make the subsequent revision.

English usage throughout the manuscript is very well and did not notice any awkward sentences or improper use of grammer.

Author Response

Dear Reviewer:

Thanks very much for your comments concerning our manuscript. The comments are all valuable and very helpful for improving our paper. As suggested, we have made corrections which we hope meet with your approval and the revisions in the manuscript have been highlighted. The main corrections in the paper and responds to the comments are as following:

The presented work on “maize genotype response to low phosphorous” is highly significant for maize breeding researchers and other peers engaged in understanding the symbiotic relationship to increase the abiotic stress adaptation for advancing the knowledge about specific genes that contribute to increased P uptake and enhanced stress adaptation. The authors have presented their work succinctly and the presented result section is highly convincing. The use of two contrasting breeding lines have clearly demonstrated the contrast differences in terms of gene expression level. Presented manuscript seems to add high value to the understanding of role of symbiotic relationship of Talaromyces purpurogenus and P absorption. I think the manuscript can be considered to accept if the Editorial Team finds it suitable.

There are other issues reported in regard to typos so please see the review comments given below.

Abstract: Abstract is written well and has rendered general significance and a synopsis of the presented research, which could be useful for a broader readership; however, it is too long (275 words) considering the 200-word limit of IJMS it needs to trim further to make it more succinct so would encourage authors to revise abstract to consider made suggestion.

Reply: Thanks for your suggestion. As suggested, we have revised the abstract to be much more succinct.

Introduction: The introduction is succinct and has been summarized very well with the relevant background of current research and challenges associated with it and how it could be addressed. Relevant literature has also been cited that supports the foundation of the presented research.

Reply: Thank you for your approval.

Results: The result section is presented very eloquently and all the findings have been presented very simply yet fully conveyed.

Reply: Thanks for your approval.

Discussion: The result findings have been very well discussed and have been corroborated with prior research and also shown how these genes impact their regulation. However, I think it would be better if authors could elaborate bit more on how these genes can be incorporated in the future breeding efforts and what it means for overall maize improvement.

Reply: Thank you for the valuable comments. As suggested, we have added more detailed information about how these genes can be incorporated in the future for maize breeding.

Materials and Methods: Authors have provided sufficient methodology details for transcriptome analysis and other mentioned analysis and peer researchers can repeat similar experiments if needed.

Reply: Thanks for your comments.

Lines 405-406: I think more details would be better concerning the statistical analysis.

Reply: As suggested, we have added more details in the statistical analysis.

Conclusion: Very well thought conclusion has been provided and Figure 8 helps to convey the findings of this research; however, Figure 8 seems to be a repetition of Figure 2D and not sure if it serves any different purpose. I would suggest that authors replace Figure 2D with Figure 8 and only mention the same in the conclusion section. Additionally, authors could also summarize the overall research goals and findings by providing other necessary key take-home messages to improve the conclusion.

Reply: Thanks for your suggestion. Figure 2D shows the photographs reflecting the root morphological appearance in –Tp and +Tp treatments. Figure 8 gives a summarize with conceptual paradigm explaining the regulatory mechanisms in different genotypes of maize during plant–T.purpurogenus symbiosis under low-P stress. Therefore, we prefer to both Figure 2D and Figure 8 in the manuscript. As suggested, we have added several sentences to improve the conclusion in the revised manuscript.

Tables and Figures:

Figure 1. There is hardly any difference in –Tp and +Tp images and it’s very hard to detect the differences especially in 31778. I think this figure go to supplementary information as it seems to be not adding a great value.

NOTE: In comparison to Figure 1, Figure 2 has clearly supports the claim that +Tp inoculation increases the shoot weight, root weight and P absorption.

Reply: Thanks for your suggestion. As suggested, we have moved Figure 1 to supplementary information.

Table 1 and 2. I think Table 1 is unnecessary to be presented in the manuscript as it is just a repetition and less relevant than the Table 1. I think Table 1 can go as a supplementary information and Table 2 can be kept in the manuscript.

Reply: As suggested, we have moved Table 1 to supplementary information.

References: Provided references are relevant and there is no unwarranted self-citation that I noticed in the provided references.

Reply: Thanks for your approval.

Thanks again for your comments and suggestions.

Best wishes!
